

# Implementation of higher-order vertical finite elements in ISSM v4.13. for improved ice sheet flow modeling over paleoclimate timescales.

Joshua K. Cuzzone[1,2], Mathieu Morlighem[1], Eric Larour[2], Nicole Schlegel[2], Helene Seroussi[2]

[1]University of California, Irvine, Department of Earth System Science, Croul Hall, Irvine, CA 92697-3100, USA
[2]Jet Propulsion Laboratory, California Institute of Technology, 4800 Oak Grove Drive MS 300-323, Pasadena, CA 91109-8099, USA

*Correspondence to*: Joshua K. Cuzzone (Joshua.K.Cuzzone@jpl.nasa.gov)

**Abstract.**

Paleoclimate proxies are being used in conjunction with ice sheet modeling experiments to determine how
the Greenland ice sheet responded to past changes, particularly during the last deglaciation. Although these comparisons have been a critical component in our understanding of the Greenland ice sheet sensitivity to past warming, they often rely on modeling experiments that favor minimizing computational expense over increased model physics. Over Paleoclimate timescales, simulating the thermal structure of the ice sheet has large implications on the modeled ice viscosity, which can feedback onto the basal sliding and ice flow.
To accurately capture the thermal field, models often require a high number of vertical layers. This is not the case for the stress balance computation, however, where a high vertical resolution is not necessary. Consequently, since stress balance and thermal equations are generally performed on the same mesh, more time is spent on the stress balance computation than is otherwise necessary. For these reasons, running a higher-order ice sheet model (e.g., Blatter-Pattyn) over timescales equivalent to the paleoclimate record has
not been possible without incurring a large computational expense. To mitigate this issue, we propose a method that can be implemented within ice sheet models, whereby the vertical interpolation along the z-axis relies on higher-order polynomials, rather than the traditional linear interpolation. This method is tested within the Ice Sheet System Model (ISSM) using quadratic and cubic finite elements for the vertical interpolation on an idealized case and a realistic Greenland configuration. A transient experiment for the
ice thickness evolution of a single dome ice sheet demonstrates improved accuracy using the higher-order vertical interpolation compared to models using the linear vertical interpolation, despite having fewer degrees of freedom. This method is also shown to improve a models ability to capture sharp thermal gradients in an ice sheet particularly close to the bed, when compared to models using a linear vertical interpolation. This is corroborated in a thermal steady-state simulation of the Greenland ice sheet using a
higher-order model. In general, we find that using a higher-order vertical interpolation decreases the need for a high number of vertical layers, while dramatically reducing model runtime for transient simulations. Results indicate that when using a higher-order vertical interpolation, runtimes for a transient ice sheet relaxation are upwards of 10 to 57 times faster than using a model which has a linear vertical interpolation, and thus requires a higher number of vertical layers to achieve a similar result in simulated ice volume,
basal temperature, and ice divide thickness. The findings suggest that this method will allow higher-order models to be used in studies investigating ice sheet behavior over paleoclimate timescales at a fraction of the computational cost than would otherwise be needed for a model using a linear vertical interpolation.

## 1 Introduction


Although the future trajectory of the Greenland ice sheet (GrIS) trends toward continued mass loss under elevated surface temperature into the future, the speed and magnitude of these changes remain unknown



(Church et al., 2013). To provide clues as to how past surface forcings influenced change over the GrIS, researchers have often relied on the paleoclimate record to serve as an analog for potential future changes (Alley et al., 2010). These records allow scientists to gain crucial insights into the evolution of the ice sheet during different climatic settings and are often corroborated by multiple lines of proxy evidence highlighting ice sheet change (e.g., ice core records, marine sediment records, terrestrial records). With respect to the GrIS, a wealth of data has been produced highlighting these changes since the beginning of the Holocene (e.g., Alley et al., 2010; Briner et al., 2016). These datasets have the potential to provide invaluable constraints for ice sheet modeling efforts aimed at exploring the sensitivity of the GrIS to past climate changes. For example, using relative sea level records throughout Greenland, Tarasov and Peltier (2002) were able to constrain an ice sheet model of the GrIS over the last deglaciation. This approach was improved through increased data coverage during later studies (Simpson et al., (2009); Lecavalier et al., 2014), highlighting the practical usage of paleoclimate proxies in ice sheet modeling efforts. Recently, ice sheet modeling results of the last deglaciation and Holocene have been compared with terrestrial records that capture changes in the ice sheet margin position (Larsen et al., 2015; Young and Briner, 2015; Sinclair et al., 2016). Because these comparisons are still relatively nascent, large model-data discrepancies do exist in some locations between the modeled margin and the margin derived from the proxy evidence, particularly in areas along the ice sheet margin where fast flow dominates. Some reasons for the model-data discrepancies include the use of a relatively coarse (10km or greater) grid and use of the shallow ice approximation (SIA; Hutter, 1983; Sinclair et al., 2016). Because the SIA was mainly developed for modeling the interior flow of ice sheets where the ice flow is dominated by vertical shear, it ignores membrane stresses (longitudinal and lateral drag) that are predominant closer to the GrIS margin (Hutter, 1983), and can lead to large thickness errors in these regions (Bueler et al., 2005). Both of these limitations have the impact of restricting how well an ice sheet model can simulate the behavior of an ice sheet near the margin, which is where the majority of paleoclimate evidence exists (Kirchner et al., 2011; Seddik et al., 2012; Seddik et al., 2016).

Nevertheless, to improve simulation speed needed for long paleoclimate spinups, ice flow models of reduced complexity often utilizing the SIA with a horizontal resolution of 10 km or greater are used to decrease computational cost, ultimately allowing for a more efficient modeling over time intervals equivalent to a glacial cycle (~120 kyr) or longer. Despite its simplification, the SIA has allowed for great strides in our understanding of the paleoclimatic evolution of the GrIS both in mass and temperature (Huybrechts, 2002; Tarasov and Peltier, 2002; Greve et al., 2011; Rogozhina et al., 2011) and its justification can be related towards its ability to sufficiently model the volume evolution of the GrIS on a scale that is consistent with the dominant flow characteristics (Furst et al., 2013). To address issues associated with the SIA, some models combine SIA and the shallow shelf approximation (SSA; MacAyeal, 1989), which allows a model to capture some of the dynamical processes occurring near ice sheet margins (Pollard and DeConto, 2009; Bueler and Brown, 2009; Aschwanden et al., 2016). To achieve this coupling however, models impose mass flux conditions at the grounding line, which serves as a boundary condition for the SSA model, or rely on tuning of a weighting parameter, whereas this discontinuity does not exist for higher-order models.

With model-data comparisons of past ice sheet changes becoming more common, however, some applications may benefit from using an ice sheet model of increased complexity, particularly when comparisons of past margin behavior is of interest. Higher-order models (Blatter, 1995; Pattyn, 2003; Hindmarsh, 2004; herein referred to as BP for Blatter-Pattyn) that include membrane stresses and elements of the vertical shear stress have been a hallmark in the ice sheet modeling community over the past decade, being favored for their ability to model both the fast and slow components of ice flow. The majority of the computational demand for an ice sheet model resides within the stress balance computation. Although the stress balance computation does not require a high vertical resolution, the thermal model usually does in order to capture sharp thermal gradients near the base of the ice. As a consequence, more runtime is needed



l00 during the stress balance computation than is necessary. Because of the increased model complexity in BP models they have therefore not been run over paleoclimate timescales due to the large computational expenses needed to complete the runs. To utilize BP models in paleoclimate simulations, methods to improve runtime speed without sacrificing the models precision need to be addressed.

l05 Here we present a method, which builds upon the thermomechanical ice flow model ISSM (Ice Sheet System Model), to improve model speed within the BP ice sheet model simulations. While our implementation and analysis are done with ISSM, the methods can be applied to a wide range of finite element ice sheet models. The main component of this development focuses on the vertical extrusion of layers within ISSM, and the type of finite elements used to create the vertical interpolation. The aim of this method is to allow the user to perform model simulations that have a smaller number of vertical layers than typically used, while still being able to precisely capture the thermal state of the ice sheet than would

l10 otherwise be captured using traditional means of linear vertical interpolation. We begin by first describing the methodology associated with the implementation of higher order vertical elements in section 2, followed by a description of the model experiment setup for an idealized single dome ice sheet and a realistic GrIS configuration in section 3. The results are accompanied by a discussion in section 4 and conclusions in section 5.

l15

## 2 Higher-order finite elements

Like many finite element ice sheet models, ISSM relies on prismatic elements, which are the result of a vertical extrusion of a 2-dimensional triangular mesh. The interpolation used in these elements is

l20 decomposed into a horizontal interpolation and a vertical interpolation. A P2xP1 finite element, for example, has a quadratic finite element on the horizontal plane (triangle) and a linear interpolation in the vertical direction. Here, we assume that the variations in model fields are accurately captured by the horizontal mesh, but that sharp gradients in the temperature at the base of the ice sheet need to be captured. For this purpose, we investigate finite elements that have three different degrees in the vertical nodal

l25 functions:1) P1 linear elements, 2) P2, with a quadratic interpolation along the z-axis, and 3) P3, with a cubic interpolation along the z-axis, as illustrated by Figure. 1.

Since the nodal functions are taken as a product of horizontal and vertical polynomials, they can be written in the following terms: $N_i(x,y,z) = f_j(x,y)g_k(z)$. Here, we keep a linear interpolation for $f_j$ and they are

l30 classically written as:

$$f_1(x,y) = x$$

$$f_2(x,y) = y \hspace{4cm} (1)$$

$$f_3(x,y) = 1 - x - y$$

l35

in the standard triangle reference element whose corners are (0,0), (1,0) and (0,1). The functions $g_k(z)$ control the degree of interpolation along the z-axis, and the nodes associated to these functions are located along the 3 vertical segments of the prism. The number of nodes along these segments depends on the degree of these polynomials.

l40

## 2.1 P1xP1 prismatic elements

In the vertical direction, we use a reference element that goes from z=-1 to z=1. A linear element (P1xP1; herein noted as P1) has 6 nodes: one per vertex. We have 6 nodal functions for the reference element, 3 in

l45 the horizontal plane (Eq.1), times 2 along the z-axis:





$$g_1(z) = \frac{1}{2}(1-z)$$

$$g_2(z) = \frac{1}{2}(1+z)$$

(2)

### 2.2 P1xP2 prismatic elements

l50

For a quadratic finite element in the vertical direction (herein noted as P2), we have 9 nodes per element (Fig. 1): one per vertex and one in the center of each vertical segment. We have the following functions in the vertical direction:

$$g_1(z) = \frac{1}{2}z(1-z)$$

l55

$$g_2(z) = \frac{1}{2}z(1+z)$$

(3)

$$g_3(z) = (1-z^2)$$

### 2.3 P1xP3 prismatic elements

l60

For a cubic finite element in the vertical direction (herein noted as P3), one needs 12 nodes per element (Fig. 1): one per vertex and 2 located at one third and two thirds of each vertical segment. The vertical components of the nodal functions are:

$$g_1(z) = -\frac{9}{16}(z-1)\left(z-\frac{1}{3}\right)\left(z+\frac{1}{3}\right)$$

l65

$$g_2(z) = \frac{9}{16}\left(z-\frac{1}{3}\right)\left(z+\frac{1}{3}\right)(z+1)$$

(4)

$$g_3(z) = \frac{27}{16}(z-1)\left(z-\frac{1}{3}\right)(z+1)$$

$$g_4(z) = -\frac{27}{16}(z-1)\left(z+\frac{1}{3}\right)(z+1)$$

### 2.4 Benefits of higher-order vertical finite elements

l70

Increasing the degree of finite elements along the z-axis is comparable to increasing the resolution along the z-axis, whereby having higher-order polynomials makes it possible to better capture sharp changes despite the number of elements in the vertical being limited to 4 or 5. Figure 2 illustrates this idea for an exponential function that is representative of a thermal profile. Here, the ice is uniformly cold throughout except at the base where the ice is warmer due to the geothermal heat flux and frictional heating. Using only 4 layers and linear elements (P1), this vertical profile is poorly captured, as the number of layers is too small to correctly represent the gradient of temperatures near the base. As quadratic elements do better, the cubic elements capture the shape of the exponential curve with maximum accuracy, even for a coarse mesh.

l80 ## 3 Model description and experimental setup





For the following model experiments we use the Ice-sheet System Model (ISSM; Larour et al., 2012), a finite element, thermomechanical ice sheet model. The tests performed in this study can be split into two experiments. We first test the precision of the higher-order vertical interpolation using a simplified single dome ice sheet experiment, following experiment A of the European Ice Sheet Modeling INiTiative (EISMINT2) experiments (Payne et al., 2000). We then apply a similar setup to a GrIS wide model, where the steady-state thermal solution is computed. Specifics regarding model setup and the relevant experiments are discussed below.

### 3.1 Single dome experiment setup

To test the performance of the higher-order vertical interpolation, we adopt a setup similar to the EISMINT2 experiments (Payne et al., 2000), which was targeted for the assessment of thermo-mechanically shallow ice models. We perform all of our experiments using a model with horizontal grid resolution of 20 km x 20 km, with a model domain of 1500 km x 1500 km. The maximum surface mass balance of 0.5 m/yr occurs at the center of the domain (over the dome summit), and linearly decreases radially as a function of the geographical distance from the dome. Accordingly, the minimum surface air temperature (238.15 K) is set at the dome summit, and decreases away from the dome following the same basis as the surface mass balance. The ice rheology is temperature dependent, following Cuffey and Paterson (2010, p. 75).

Rather than performing all of the experiments associated with the EISMINT2 benchmarks, we choose to limit the analysis to only experiment A. Experiments begin with zero ice over a bed with flat topography and are run to relaxation for 100,000 years. To compare the differences between the vertical interpolations, we run 24 simulations in total. These simulations use the P1, P2, and P3 vertical interpolation for models that have a minimum of 3 non-uniform layers to a maximum of 10 non-uniform layers. Each model uses an extrusion exponent of 1.2, indicating that the layers are not equally spaced but rather modestly biased towards thinner layers at the bed. Aside from comparison of the results to EISMINT2, we run a simulation using the P1 vertical interpolation on a model with 25 layers. This model will serve as the benchmark to compare the other simulations to, with a 25 layer P1 model being representative of what is typically used in the setup for GrIS wide simulations in ISSM (Seroussi et al., 2013). We note that for the stress balance computation we use the P1 vertical interpolation, while the thermal computation makes use of the higher order vertical elements.

### 3.2 GrIS model setup

In addition to comparison with the EISMINT2 experiment A, thermal steady-state computations are performed for a GrIS wide model to determine how well the vertical interpolations can capture thermal profiles and basal temperatures throughout the ice sheet. The three-dimensional higher-order model (i.e. BP) of Blatter (1995) and Pattyn (2003) is used for the momentum balance equations. The nonlinear effective ice viscosity result from Glen's flow law (Glen, 1955) and is given in equation 4 of Larour et al. (2012). The ice hardness, B, is temperature dependent following the rate factors given in Cuffey and Paterson (2010, p. 75), while the basal drag is empirically determined following a viscous flow law outlined in Cuffey and Paterson (2010).

The GrIS wide model relies on anisotropic mesh adaptation, whereby the element size is refined as a function of surface elevation (Howat et al., 2014) and InSAR surface velocities from Rignot and Mouginot (2012), becoming finer in areas of steep topography and large ice flow gradients. The model mesh has a horizontal resolution ranging from 3 km in areas of ice streams to 20 km over the interior regions where the ice flow is slow, corresponding to a two-dimensional model with ~10,000 triangular elements. The



horizontal mesh is then extruded to the corresponding number of layers outlined in section 3.1. This results in 24 models with a 3-D mesh ranging from 30,000 to 100,000 prismatic elements, depending on the models number of vertical elements. Similar to the experiments outlined in section 3.1, we run a benchmark thermal steady-state simulation using a model that has 25 non-uniform layers and uses the P1 vertical interpolation (250,000 elements).

The models are initialized with bed topography from BedMachine Greenland v3 (Morlighem et al., 2017), and ice surface elevation from the GMIP DEM of Howat et al. (2014). The surface mass balance and surface temperatures are taken from Ettema et al. (2009), and the geothermal heat flux relies on a setup identical to Seroussi et al. (2013). The underlying geothermal heat flux from Shapiro and Ritzwoller (2004) is used, however, values of 20 mWm$^{-2}$ and 60 mWm$^{-2}$ are added at the Dye3 and GRIP sites respectively, after Seroussi et al. (2013). These modifications follow an exponential decay from the particular sites with a radius of 250 km.

The thermal model for both the single dome and steady-state experiments use an enthalpy formulation derived from Aschwanden et al. (2012), which includes both temperate and cold ice. At the ice surface, air temperature is imposed, while the geothermal heat flux is applied at the base. For full details outlining the thermal model used in ISSM we direct the reader to Seroussi et al. (2013) and Larour et al. (2012). Lastly, the spatially varying basal drag coefficient is determined using inverse methods (Morlighem et al., 2010; Larour et al., 2012), providing the best match between modeled and InSAR surface velocities from Rignot and Mouginot (2012).

## 4 Results and discussion

### 4.1 Single dome experiment

Each individual model is relaxed for 100,000 years to bring the ice sheet into steady-state both with respect to the ice thickness and temperature. In Fig. 3, the ice volume for each particular simulation is shown as a percent difference from the 25 layer P1 simulation with the shading corresponding to the zone where models fall within 2% of the ending ice volume simulated by the 25 layer P1 model. Although all models simulate the same relative trend for the ice volume relaxation, they do not all converge on the ice volume simulated by the 25 layer P1 model. For the models where the linear (P1) interpolation (Fig. 3A) is used in the thermal model, only those models with at least 8 layers and above fall within the 2% range of ending ice volume for the 25 layer P1 simulation. When using a higher-order vertical interpolation (P2 and P3), however, models with 4 layers and above fall within the 2% range (Figs. 3B and 3C).

To further compare the performance of each model, the corresponding ice volume, ice divide basal temperature, and ice divide thickness are shown in Table 1 for each model simulation and are compared to the mean values derived from the EISMINT2 experiment A results (Payne et al., 2000). Similar to Rutt et al. (2009), we compare our simulated values to the mean and the standard deviation of the values for experiment A in the EISMINT2 experiment. In general, models using the higher-order vertical interpolation tend to better match the EISMINT2 results. Models with 4 layers or more using the P2 or P3 vertical interpolation fall within 1 standard deviation (σ) of the mean for simulated ice volume, whereas models using the linear vertical interpolation require 8 or more layers to satisfy this constraint. With respect to the basal temperatures simulated at the ice divide, only the 10 layer P2, 10 layer P3, and the 25 layer P1 simulations fall within 1 σ of the mean for the EISMINT2 experiment A results.

Models with 5 or more layers using the P2 or P3 vertical interpolation fall within 2 σ of the EISMINT2 experiment A mean for basal temperatures simulated at the ice divide, while at least 7 layers are needed for models using the linear vertical interpolation. Regarding ice divide thickness, none of the models with 10



layers or less using the linear interpolation fall within 3 σ of the mean, however, the 25 layer P1 simulation does. Generally, models using at least 6 layers and the P2 or P3 vertical interpolation fall within at least 3 σ of the mean for the simulated ice divide thickness. Interestingly, whereas the P3 models with 6 layers and above only fall within 3 σ of the mean, models with 8 layers and above for the P2 interpolation fall within 2 σ of the mean. This is likely explained by the slightly higher temperatures simulated with the P2 interpolation, which may feed back onto the ice rheology and correspondingly, the ice flow. We note however that these differences are small, and overall models using the P2 and P3 vertical interpolation show excellent agreement amongst each other. From this exercise, it can be concluded that when using fewer layers, models that utilize the higher-order vertical interpolation are more capable of capturing the simulated ice volume, ice divide basal temperatures and ice divide thickness simulated by the EISMINT2 experiment A models. Although some differences do exist between our simulated values and those derived from the EISMINT2 experiment A results, the precision of the models using the P2 or P3 vertical interpolation is reasonable. As noted by Rutt et al. (2009), there are inherent difficulties in associating particular differences to specific model processes. Most differences in the simulated temperature can have feedbacks on the ice rheology and therefore the ice flow, which make comparisons with models using different discretization methods difficult. Overall, comparison with the EISMINT2 experiment A results demonstrate that by using fewer layers with a higher-order vertical interpolation, models are capable of capturing particular constraints more accurately than would otherwise be simulated using a linear vertical interpolation.

Because of the potential difficulties in assessing differences between our results and those derived from the EISMINT2 experiment A, we also compare our results to the model simulation using the 25 layer P1 vertical interpolation. Because this model is representative of what is characteristically used for three-dimensional, thermomechanical modeling in ISSM (Seroussi et al., 2013), further comparisons can be made to those models that agree well with simulated ice volume, ice divide basal temperature, and ice divide thickness from the 25 layer P1 model. In Table 2, the absolute value of the percent difference is shown between each individual model simulation and that using the 25 layer P1 model. Following from the comparison with the EISMINT2 experiment A results, the higher-order vertical interpolation allows models with fewer layers to capture changes simulated by the 25 layer P1 model with a higher precision. In Table 2, the green shading denotes those model simulations where the simulated ice volume, ice divide basal temperature, or ice divide thickness is within 1% of the 25 layer P1 model. Generally, models with at least 4 (P3) and 5 (P2) layers capture the simulated ice volume within 1% of that simulated by the 25 layer P1 model. Using the linear vertical interpolation, 10 layers are needed before simulating ice volume within 1% of the 25 layer P1 model. This is better illustrated in Fig. 4, where the percent difference in ice volume from the 25 layer P1 model is shown as a function of the number of layers in each model. Those models using the P2 and P3 vertical interpolation converge significantly faster to ~0-1% difference at 4-5 layers from the 25 layer P1 model. We note that the negative difference for the P2 and P3 models arises as the temperatures simulated with the higher-order vertical interpolation are slightly higher, but not significantly different than that simulated by the 25 layer P1 model (Table 2), providing a feedback between ice rheology and ice flow. Lastly, the ice divide thickness follows a similar trend in that using the higher-order vertical interpolation allows a model with fewer layers to capture what is simulated with the 25 layer P1 model (Table 2). When viewed as ice profiles extending from the dome summit to the ice edge for 3, 5, and 7 layer models (Fig. S1), the differences in ice thickness between models appear small, with the P2 and P3 being almost identical, and only minor differences existing for the models using the P1 vertical interpolation.

Differences between the linear vertical interpolation and the P2 or P3 interpolation become more apparent when analyzing ice temperature profiles. In Fig. 5, ice temperature profiles are plotted at the ice divide for models with 3, 5, and 7 layers. With only 3 layers, models with the P1, P2, and P3 vertical interpolation simulate a temperature profile that is too warm between 500 to 1500 meters, and too cold approaching the





base. Despite the vertical interpolation used, the profile is not well captured, although improvements to the shape of the temperature profile in the transition between 500 to 1500 meters can be seen in models using the higher-order vertical interpolation. Adding more layers to each model improves the overall fit to the 25 layer P1 model, although the models using the P2 and P3 vertical interpolation capture the shape of the temperature profile much better than the linear interpolation. The overall fit is improved not only at the base but also in the transition between 500 to 1500 meters where the ice begins to warm more rapidly approaching the base. We also find that the differences between the P2 and P3 vertical interpolation are marginal in this example, indicating that using a quadratic vertical interpolation (P2) is suitable when given the choice to using a cubic vertical interpolation (P3).

## 4.2 Improvements in simulation speed

Although much of the success regarding the higher-order vertical interpolation resides in the models ability to capture the vertical structure of temperature in the ice using fewer layers than is needed from the traditional linear vertical interpolation, improvements to model speed are the main motivation for its implementation, particularly in BP models. To test how model speed is improved when implementing the higher-order vertical interpolation, we begin by using the relaxed model simulations that have thus far only used the shallow ice approximation. From the relaxed model states, each simulation is run for 100 years using the BP ice flow model in ISSM, and uses the same boundary conditions from the relaxation with a fixed timestep of 0.2 years.

Since we assume that the horizontal mesh accurately captures variations in the model fields, running a higher-order vertical interpolation reduces the number of layers used in the stress balance computation, which is the most computationally demanding part of transient simulations. Comparing the simulation time for each individual model compared to the 25 layer P1 model, all models, despite the vertical interpolation used, complete the 100 year run anywhere between 241 (3P1) to 9 (10P3) times faster (Fig. 6). To determine how models perform based upon the vertical interpolation, a criteria is established based upon Table 2, such that each models simulated ice volume must be within 1% of those values simulated by the 25 layer P1 model, which represents the relative uncertainty associated with the present day ice volume of the GrIS (Morlighem et al., 2017). Based upon these criteria, models using the P1 vertical interpolation must have 10 layers or more, while models using the P2 and P3 vertical interpolation can use at least 5 or 4 layers respectively. When applying these criteria, runtime is 5 times faster for a 5 layer P2 model versus a 10 layers P1 model. If we assume a 7 layer P1 model is adequate, the runtime for a 5 layer P2 model is 2 times faster. When compared with the 25 layer P1 model, the 5 layer P2 model completes the relaxation 57 times faster.

## 4.3 Application to a GrIS wide model

The thermal steady state simulation is compared with the GRIP ice core record (Dahl-Jensen et al., 1998) in Fig. 7 for models with 3, 5, and 7 layers as well as the 25 layer model with the P1 vertical interpolation. The simulated thermal structure for the 25 layer P1 model is similar to the thermal profile presented in Seroussi et al. (2013). Temperature differences of 2-5 degrees occur between the models and the GRIP record between 1200 to 2200 meters, and 500 to 1000 meters, however, this is consistent with other models computing the thermal steady-state (Dahl-Jensen et al., 1998; Rogozhina et al., 2011). The influence of past surface temperatures, ice flow history, and accumulation are not represented in our thermal steady-state computation. Spinning up an ice sheet model over a glacial cycle typically provides a better match to the ice core records but is beyond the scope of this experiment (Greve, 1997; Rogozhina et al., 2011). Nevertheless, the general profile is well simulated, with only minor differences in the simulated basal temperatures for the models using P2 or P3 interpolations. Similar to the results presented for the ice dome (Fig. 5), models using the higher-order vertical interpolation simulate the shape of the thermal profile





(compared to 25 layer P1) much better than the models using the linear vertical interpolation and the same number of layers. When examined spatially, the difference in basal temperature decreases using a model with a higher-order vertical interpolation, particularly over the interior of the ice sheet (Fig. S2a-c). Although differences between models using the P1 vertical interpolation and the 25 layer model begin to
minimize with 8 layers, the differences for models using the P2 and P3 vertical interpolation become small with 4-5 layers.

## 5 Conclusion

This study aims at addressing the current computational limitation in using higher-order stress balance ice sheet models for paleoclimate studies. Currently, analysis of ice sheet modeling experiments focusing on the past behavior of the GrIS are being complemented with rich paleoclimate data constraining features of the past ice sheet behavior (Larsen et al., 2015; Young and Briner, 2015; Sinclair et al., 2016). Where shallow ice models might be limited in their ability to simulate the marginal behavior of the GrIS through
the exclusion of higher-order stress terms and an inability to run on a high-resolution mesh, BP models may become more appropriate for such comparisons in the future. To help alleviate the computational expense in using a BP model, we implement higher order vertical elements. As shown in section 4.1 of this study, increasing the degree of the vertical interpolation allows the model to capture gradients in the thermal profile of the ice with more precision than would otherwise be captured using a model with a linear vertical
interpolation, despite having the same number of vertical layers. Models with correspondingly fewer layers that used the higher-order vertical interpolation were able to capture the transient behavior consistent with the EISMINT2 experiment A results (Payne et al., 2000) and also performed well when compared to a model similar to those that are used for modeling studies in ISSM (Seroussi et al., 2013).

The biggest attraction for using higher order vertical elements is that they not only decrease the computational burden for the thermal model, but also for the stress balance computation, due to a decrease in the number of vertical layers needed. Overall, this leads to a large reduction in computational time, particularly when a BP model is used. Models using the higher-order vertical interpolation were shown to shorten runtime anywhere between 2 to 5 times for a 5 layer model compared to models with 7 and 10
layers respectively, using a linear vertical interpolation. When compared to the 25 layer model using the linear vertical interpolation, models with 5 to 10 layers using the higher-order vertical interpolation had anywhere between a 57 to 10 times faster runtime, with minimal impacts on the precision of the simulated ice volume and thermal state. When the higher order vertical elements were applied to a 3 dimensional, BP model of the GrIS, experiments showed the thermal state of the ice sheet can be captured as precisely as
our 25 layer P1 model when at least 5 layers are used for a quadratic (P2) vertical interpolation and at least 4 layers for a cubic (P3) vertical interpolation. When comparing the quadratic and cubic vertical interpolation, the benefits of using a cubic vertical interpolation are slight, although it may be useful when modeling in areas of complex thermal regimes.

In the context of paleoclimate simulations, using a higher-order vertical interpolation improves simulation speed, particularly for BP ice sheet models. BP models using this will still likely be too computationally intensive for simulations which sample parameter space and thus require multiple independent simulations (Applegate et al., 2012; Robinson et al., 2011). However, in experiments where BP models may offer improvements in model data comparison versus using shallow ice models, higher-order vertical elements
can be used as a means to improve model speed while still being able to capture the qualities simulated in a model with many more layers, but at the fraction of the speed. In this respect, future studies will use these higher-order vertical elements to enhance computational speed while maintaining mechanical complexity for ice sheet modeling experiments over various paleoclimate timescales.






## Code availability

The higher order finite elements are currently implemented in the ISSM code, which can be compiled
following the instructions on the ISSM website (https://issm.jpl.nasa.gov/download).

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

**Tables**


| | Volume ( $10^6$ km$^3$) | Ice divide basal temp (K) | Ice divide thickness (m) |
|---|---|---|---|
| **Eismint 2 exp. A (mean value)** | | | |
| *Payne et al., 2000* | **2.128 ± 0.051** | **255.605 ± 1.037** | **3688.3 ± 27.757** |
| 25 layer P1 | 2.144 | 254.723 | 3767.0 |
| 3 layer P1 | 2.344 | 247.229 | 4093.2 |
| 4 layer P1 | 2.265 | 250.240 | 3960.4 |
| 5 layer P1 | 2.231 | 252.351 | 3876.5 |
| 6 layer P1 | 2.209 | 253.285 | 3844.4 |
| 7 layer P1 | 2.192 | 253.793 | 3823.0 |
| 8 layer P1 | 2.179 | 254.115 | 3806.7 |
| 9 layer P1 | 2.171 | 254.337 | 3794.5 |
| 10 layer P1 | 2.165 | 254.480 | 3785.4 |
| 3 layer P2 | 2.264 | 249.873 | 4023.2 |
| 4 layer P2 | 2.169 | 252.598 | 3838.1 |
| 5 layer P2 | 2.146 | 253.717 | 3785.8 |
| 6 layer P2 | 2.138 | 254.225 | 3764.8 |
| 7 layer P2 | 2.131 | 254.488 | 3753.9 |
| 8 layer P2 | 2.124 | 254.532 | 3747.1 |
| 9 layer P2 | 2.123 | 254.634 | 3743.6 |
| 10 layer P2 | 2.122 | 254.656 | 3741.3 |
| 3 layer P3 | 2.245 | 250.019 | 4002.0 |
| 4 layer P3 | 2.160 | 252.689 | 3826.4 |
| 5 layer P3 | 2.145 | 253.581 | 3779.3 |
| 6 layer P3 | 2.143 | 253.895 | 3765.0 |
| 7 layer P3 | 2.138 | 254.213 | 3756.5 |
| 8 layer P3 | 2.131 | 254.334 | 3750.3 |
| 9 layer P3 | 2.129 | 254.436 | 3748.5 |
| 10 layer P3 | 2.127 | 254.600 | 3746.2 |

Table I. Ice volume, ice divide basal temperature, and ice divide thickness for each individual simulation after a 100 kyr relaxation. Also shown are the corresponding mean values for the EISMINT2 (Payne et al., 2000) experiment A simulation and the standard deviation. The shading indicates those simulations whose values fall within 1 standard deviation (green), 2 standard deviations (blue,) and 3 standard deviations (red) from the EISMINT2 experiment A mean values.



| | Volume ( $10^6$ km³) | Ice divide basal temp (K) | Ice divide thickness (m) |
|---|---|---|---|
| 3 layer P1 | 9.33 | 2.94 | 8.66 |
| 4 layer P1 | 5.64 | 1.76 | 5.13 |
| 5 layer P1 | 4.06 | 0.93 | 2.91 |
| 6 layer P1 | 3.03 | 0.56 | 2.05 |
| 7 layer P1 | 2.24 | 0.37 | 1.49 |
| 8 layer P1 | 1.63 | 0.24 | 1.05 |
| 9 layer P1 | 1.26 | 0.15 | 0.73 |
| 10 layer P1 | 0.98 | 0.10 | 0.49 |
| 3 layer P2 | 5.60 | 1.90 | 6.80 |
| 4 layer P2 | 1.17 | 0.83 | 1.89 |
| 5 layer P2 | 0.09 | 0.39 | 0.50 |
| 6 layer P2 | 0.28 | 0.20 | 0.06 |
| 7 layer P2 | 0.61 | 0.09 | 0.35 |
| 8 layer P2 | 0.93 | 0.08 | 0.53 |
| 9 layer P2 | 0.95 | 0.04 | 0.62 |
| 10 layer P2 | 0.98 | 0.03 | 0.68 |
| 3 layer P3 | 4.71 | 1.85 | 6.24 |
| 4 layer P3 | 0.75 | 0.80 | 1.58 |
| 5 layer P3 | 0.05 | 0.45 | 0.33 |
| 6 layer P3 | 0.05 | 0.33 | 0.05 |
| 7 layer P3 | 0.28 | 0.20 | 0.28 |
| 8 layer P3 | 0.61 | 0.15 | 0.44 |
| 9 layer P3 | 0.70 | 0.11 | 0.49 |
| 10 layer P3 | 0.79 | 0.05 | 0.55 |


Table II. The absolute value of the percent difference between each individual model run and the 25 layer P1 simulation at the end of the 100,000 year relaxation for ice volume, ice divide basal temperature, and ice divide thickness Green shading denotes models that fall within 1% of the variables simulated by the 25 layer P1 model at the end of the relaxation.






**Figures**

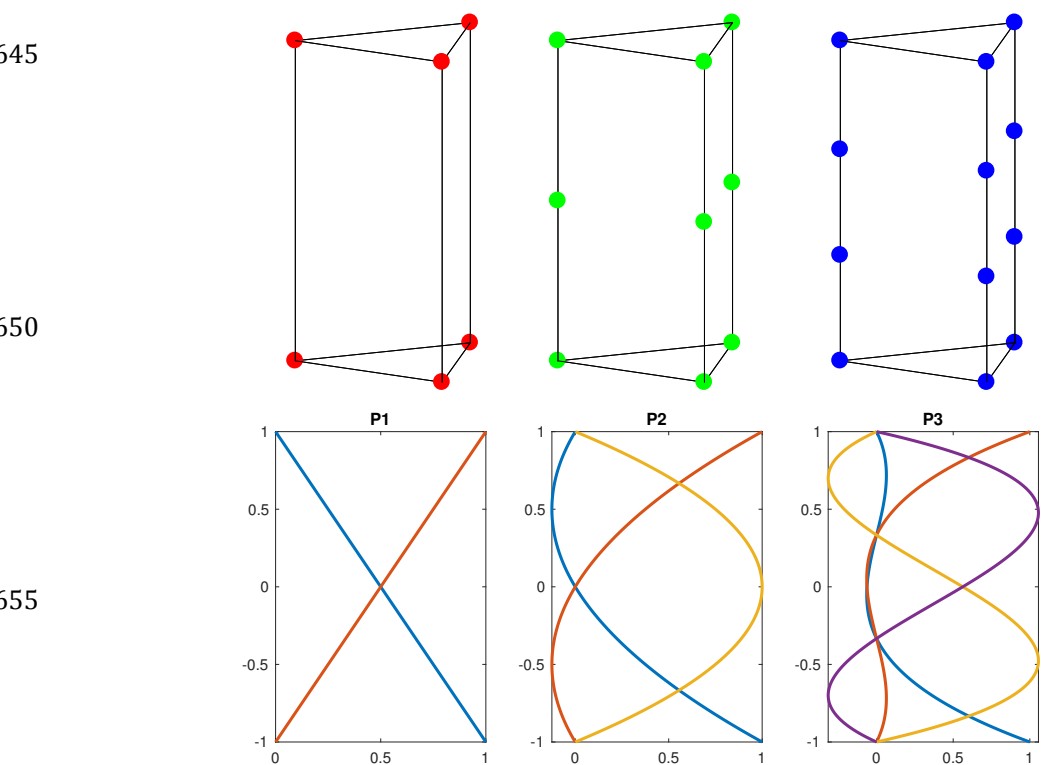




Figure 1. Top row: nodes for the P1×P1, P1×P2, and P1×P3 prismatic finite element, respectively. Bottom
row: vertical nodal functions for P1, P2 and P3 finite elements.



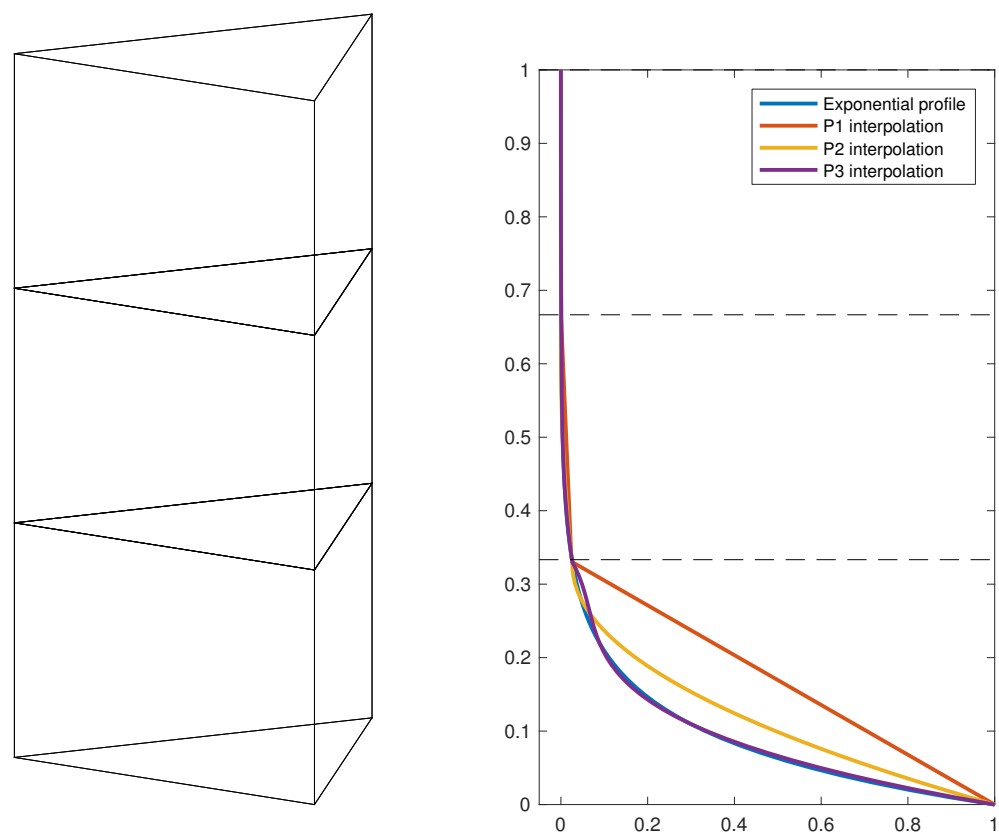

Figure 2. On the left is an example of P1×P3 prismatic elements. On the right is an example of exponential profile captured by P1, P2 and P3 finite elements. With higher order finite elements in the vertical, sharp 585 gradients in temperature are captured more precisely than with a linear (P1) interpolation.











Figure 3. The percent difference in ice volume from the 25 layer P1 model for models using the P1 (a), P2 (b), and P3 (c) vertical interpolation scheme over the 100,000 year relaxation. The gray shading denotes models that fall within 2% of the simulated ice volume for the 25 layer P1 model at the end of the 100,000 year relaxation. Only those models that fall within 2% of the simulated ice volume for the 25 layer P1 model are labeled.







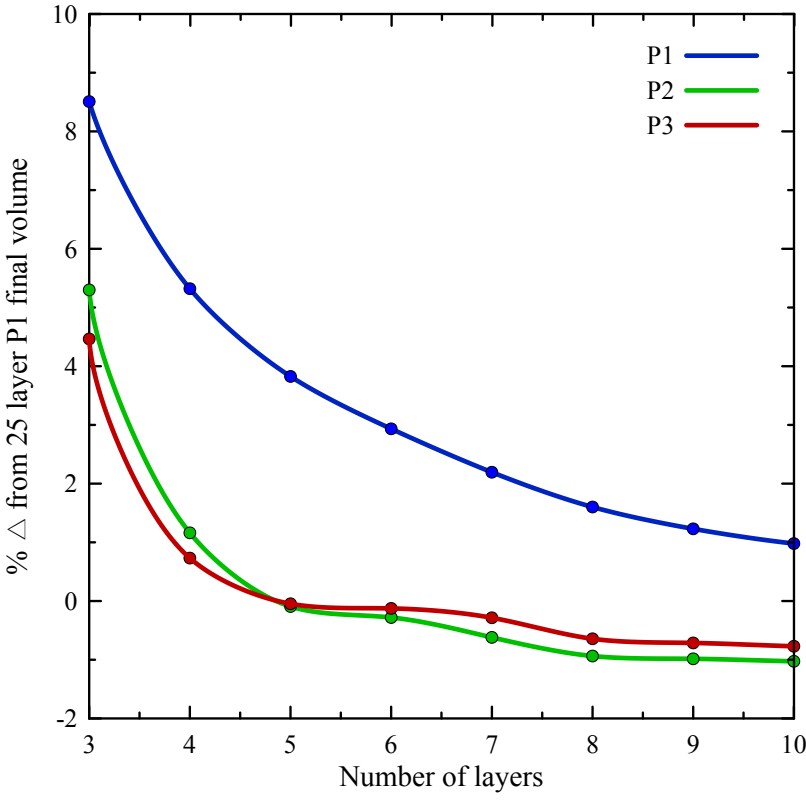


Figure 4. The percent difference in simulated ice volume after the 100,000 year relaxation for the single ice
dome experiment compared to the 25 layer P1 model. Each model run is shown as a function of the
vertical interpolation and the number of layers used.








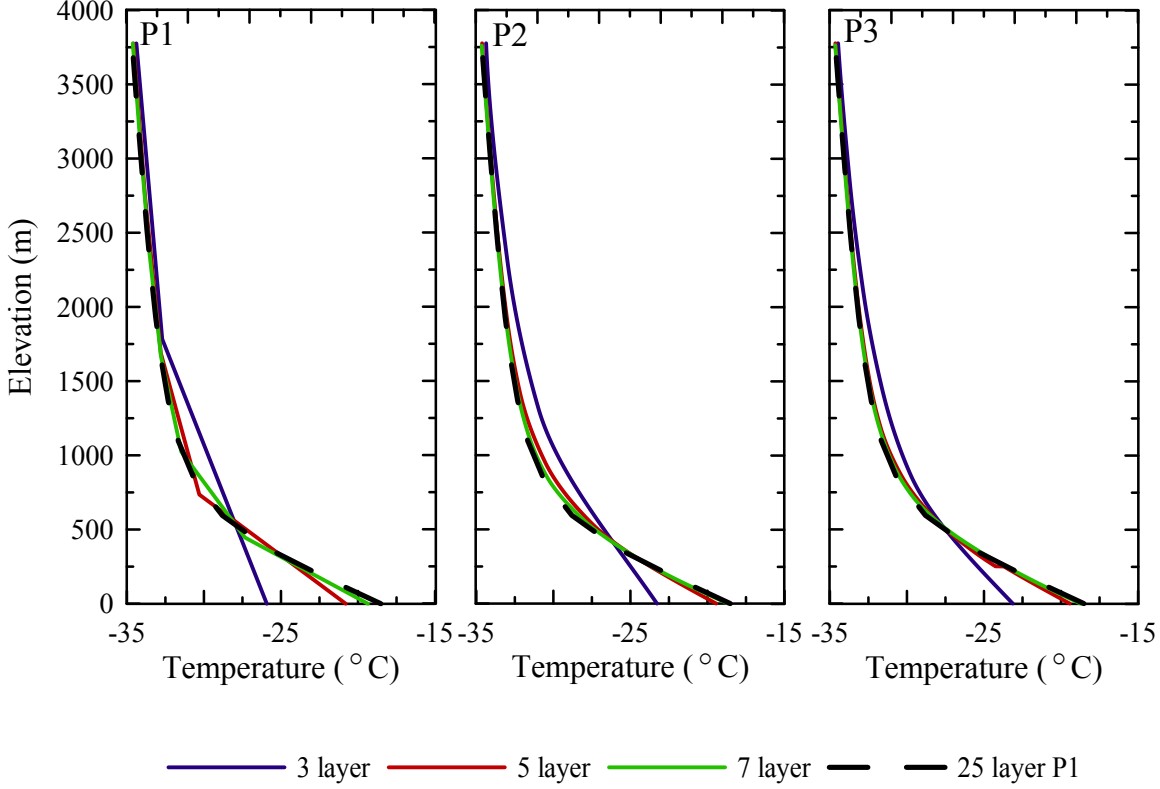

Figure 5. The resulting temperature profiles at the ice divide after the 100,000 year single ice dome
relaxation for models with 3, 5, and 7 layers, compared to the temperature profile from the 25 layer P1
model.







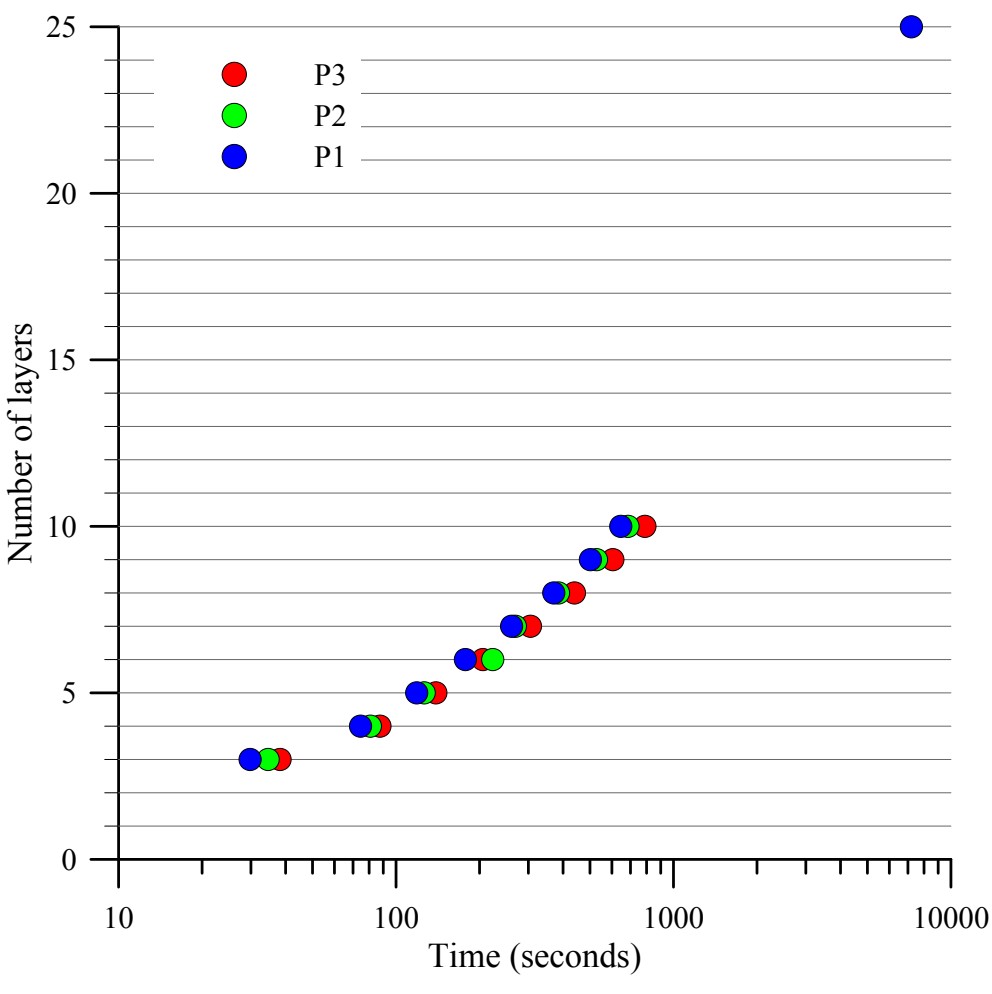

Figure 6. Run times for the 100 year higher order simulation of the single ice dome for each individual model based upon the number of layers and vertical interpolation scheme used.





305

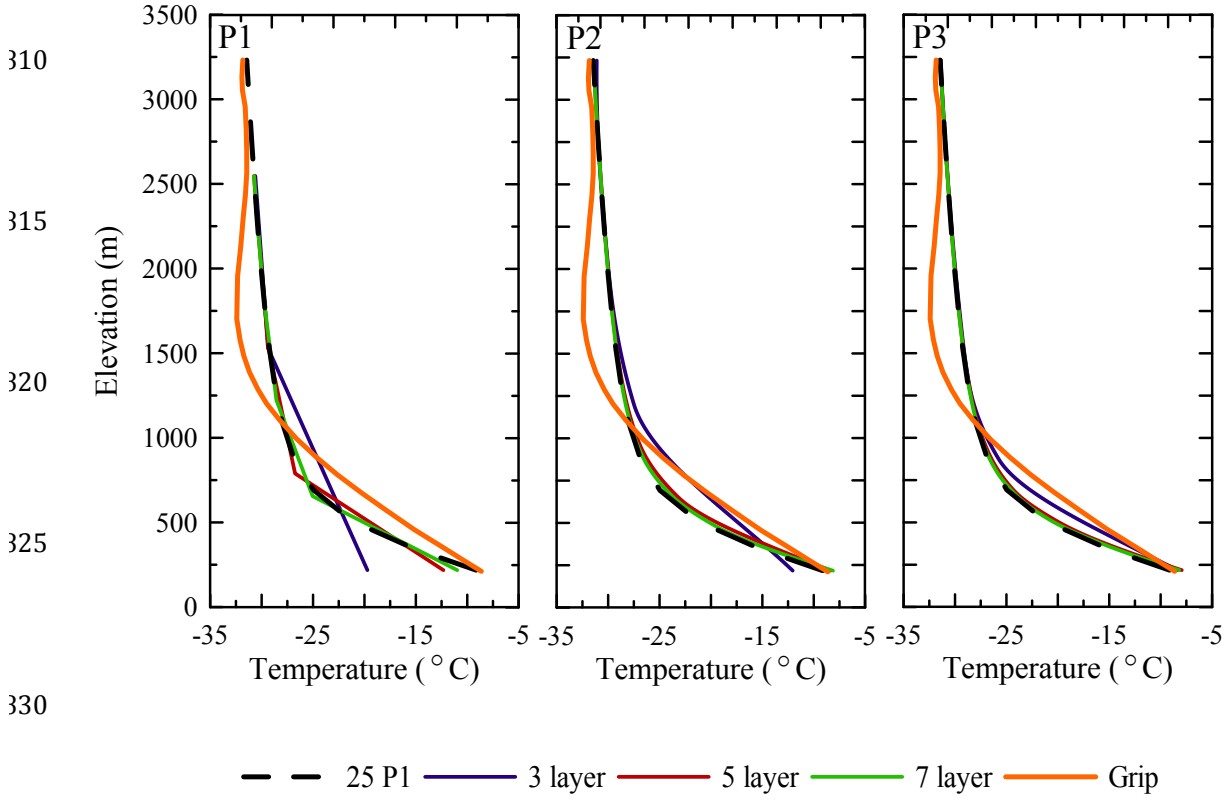

Figure 7. The resulting temperature profiles for the higher order steady state thermal computation at
335 the GRIP ice core site location for models with 3, 5, and 7 layers, compared to the temperature profile
from the 25 layer P1 model and the measured GRIP temperature profile (Dahl-Jensen et al., 1998).
