# Peer review of "Implementation of higher-order vertical finite elements in ISSM v4.13 for improved ice sheet flow modeling over paleoclimate timescales"

_Geoscientific Model Development, 2017_

## Referee Comment (RC1) · Anonymous Referee #1 · 17 Feb 2018

This paper compares the performances of a finite element thermo-mechanical ice flow model, depending on the polynomial approximation in the vertical direction. The paper mainly focus on the ability of the model to capture the sharp vertical gradients of the temperature near the ice-sheet base. They show that, for the same element size, convergence to a reference solution is faster with quadratic or cubic elements compared to linear elements. The conclusion is that, runtimes can be improved by one order of magnitude when using higher-order elements compared to linear elements, for a similar accuracy.

[Figure]

It is well known that the convergence of the finite element method with respect to the element size depends on the polynomial approximation, so that the results are not surprising. However, I think that it has never been discussed in the context of ice flow modelling, making this discussion relatively interesting for the ice flow modelling community.

Few points deserve more consideration:

- Abstract: the authors report runtimes 10 to 57 times faster. I think this is a bit misleading, as these numbers are when comparing the higher order elements with the reference 25 layers-P1 elements simulation. Improvement in runtimes are interesting if they allow to reach a similar accuracy. As the 25 layers-P1 elements simulation is used as a reference, the respective accuracy of the solutions is not known. However, it is shown that compared to the same reference, a 10 layers-P1 elements simulation falls within the same criteria of $1\%$. Improvement in runtimes is then only a factor 5. I think this is the more correct number to report in the abstract.

- Page 2 , lines 95-96: it is said that the stress balance requires less vertical resolution. I'm not sure that this is a well established result, as for areas with high friction near the base, there is also very sharp gradient of the stresses and strain rates, also requiring higher resolution for the stress balance.

- Sec. 2.4. Figure 2 compares an exponential function captured by vertical elements with different polynomial interpolation. We understand that the figure is for 1 P3 element (i.e. 4 layers of nodes) or 3 P1 elements (i.e. 4 layers of nodes); but what is the corresponding number of P2 elements?

- Sec. 3.1: it is said that P1 elements are used for the stress balance. What is the default number of integration points. Does it allow to capture the temperature profile, affecting the viscosity, within the element?

- Sec. 3.1. All the introduction is about using higher order models for the stress balance, however most of the EISMINT comparison is done with the Shallow ice model, and we learn this very late in the results section. It should be said here that the 100 000 years rexperiment is done with the SIA, justifying the comparison with EISMINT2, and that the BP model is used only to do 100 years relaxations.

- Page 5, line 227. It is said that the elements are finer in areas of steep topography and ice flow gradients. I think that the refinement is based on the second derivatives, not the gradients, so elements are finer where changes in slope and ice flow gradients are high?

---

## Referee Comment (RC2) · Anonymous Referee #2 · 22 Feb 2018

General comments:

Projections of future ice-sheet contributions to sea level rise require models that include the best possible physics and capture the effects of prior ice-sheet evolution. To this end, "spin-ups" over glacial cycles constrained by paleoclimate data are very useful. However, computational demands have limited these long simulations to models with simplified physics and parameterization. Some of these models have produced excellent results (e.g., several studies of Antarctica by Pollard and DeConto) but it is still highly desirable to incorporate higher-order physics into simulations over paleoclimate

timescales.

This study presents highly promising results by applying quadratic and cubic finite elements to discretize the vertical dimension of the Ice Sheet System Model (ISSM). Resolution of thermal gradients with linear vertical elements requires many layers, which can cause unnecessary computational expense when the momentum balance is solved on the same grid. The authors show that the use of higher degree vertical elements requires many fewer levels and results in roughly an order of magnitude improvement in computational speed for an idealized experiment.

Specific comments:

Introduction) It would be helpful to expand on the descriptions of the types of physics in order to more strongly make the case that Blatter-Pattyn is both desirable and expensive. I don't think you need any equations, but the hierarchy could be more explicit in terms of physical assumptions and their mathematical/numerical consequences. For a modeling journal like GMD, more detail than for a "regular" journal seems appropriate.

Line 96) "stress balance computation does not require a high vertical resolution" could use a reference.

Section 2) This could use a reference to your favorite finite element method book for the definitions of the elements.

201) Totally OK to only run one of the EISMINT2 experiments, but explain very briefly why you picked A. Presumably because it's the initial spin-up?

Section 3.1) Although you do say you're comparing to an experiment for SIA models, it should be more clear in the text that you're using SIA.

Section 3.2, first paragraph) It should be more clear what your thermal steady-state computation is, i.e. a simultaneous solution of heat transport and momentum equations coupled by viscosity. Also, the description of the effective viscosity can probably just say it's a function of strain rates/velocity gradients and a temperature dependent

hardness. The variable name B isn't necessary when the equation isn't shown.

Section 4.1) It's worth pointing out that Experiment A is a comparison of models without a known analytic solution, some of which produced more plausible (to me, anyway) solutions than others (e.g., W, X and Y). Comparison to the EISMINT2 results that don't show stability problems may be more reasonable. Also, it seems possible that your 25-layer results are better than the EISMINT2 models.

Section 4.2) Given that a major selling point of the new method is making BP affordable, I'd like to see some direct description of the dynamical results at some point. How do your calculated velocities compare for different numbers of layers and vertical elements? Is vertical shearing captured well? You only discuss experiments for which SIA is reasonable. I have some concern about whether the reduction in the number of levels will work for transient runs with more realistic geometry than the ice dome that include areas in which BP is necessary.

347) "we began by using the relaxed model simulations that have thus far only used the shallow ice approximation..." Please be a little more clear that you mean the single-dome calculations from 3.1/4.1.

Technical corrections:

76) cut "a"

77) cut "for"

80) "to" instead of "towards"

109) add "more" for comparison – "to more precisely capture"

129) This formula could use a space or a multiplication sign so that the factors don't run together, which is hard to read.

143) Space on either side of the =

177) "While" instead of "As"

192) "were" instead of "was", "thermomechanical"

256) "with respect to both"

263) cut either "at least" or "and above"

357) "criterion"

Table 2) The % values should be unitless.

Figure 3) I can easily figure out what you mean, but probably good to define the labels.

---

## Author Comment (AC1) · 5 Apr 2018

Reviewer #1 comments and response:

We thank the reviewer for his/her helpful comments and corrections. Below are our responses to the particular comments and corrections.

**Comment 1**: "Abstract: the authors report runtimes 10 to 57 times faster. I think this is a bit misleading, as these numbers are when comparing the higher order elements with the reference 25 layers-P1 elements simulation. Improvement in runtimes are interesting if they allow to reach a similar accuracy. As the 25 layers-P1 elements simulation is used as a reference, the respective accuracy of the solutions is not known. However, it is shown that compared to the same reference, a 10 layers-P1 elements simulation falls within the same criteria of 1%. Improvement in runtimes is then only a factor 5. I think this is the more correct number to report in the abstract."

**Response:** We agree with the reviewer that the stated runtime speed increases can be misleading in this context. In section 4.2 we highlight the associated runtime speed increases between the 25 layer P1 model and the 4/5 layer P2/P3 models, which is referenced in the abstract. Following the reviewer suggestion, and because we deem the 10 layer P1 model to fall within the criteria set in section 4.2, the abstract would be better suited to associate runtime speed increases between the 4/5 layer P2/P3 model and the 10 layer P1 model. When doing so, the runtime speed increase between a 5 layer P3/4layer P2 and a 10 layer P1 model are 5 to 7 times faster. Considering these changes, we updated the abstract accordingly.

"Results indicate that when using a higher-order vertical interpolation, runtimes for a transient ice sheet relaxation are upwards of 5 to 7 times faster than using a model which has a linear vertical interpolation, and thus requires a higher number of vertical layers to achieve a similar result in simulated ice volume, basal temperature, and ice divide thickness."

**Comment 2**: "Page 2 , lines 95-96: it is said that the stress balance requires less vertical resolution. I'm not sure that this is a well-established result, as for areas with high friction near the base, there is also very sharp gradient of the stresses and strain rates, also requiring higher resolution for the stress balance."

**Response**: We agree that this is not a well-established result in the literature. In our experience with ISSM, the stress balance is not as sensitive to changes in vertical resolution compared with the thermal computation. To test how sensitive the stress balance computation was in our GrIS model to changes in vertical resolution, we began by collapsing our 25 layer P1 model. By collapsing the 25 layer model, the ice viscosity parameter (B) is depth averaged, and therefore does not depend on depth. We next extruded our collapsed model to 25 layers and 5 layers, and ran a stress balance computation.

We have attached the results of that experiment (Figure 1. A, B, C). When comparing the stress balance surface velocity differences (A) between the 25 layer and 5 layer model, the area averaged difference is 0.22%, with the maximum difference being 6.2%. The area averaged difference in basal velocities (B) are 0.012%, with a maximum difference of 2.6%. Lastly, the area averaged difference in basal shear (C) is 0.54 m/yr with a maximum of 87 m/yr. From this

experiment we conclude that the differences associated with the vertical resolution in the stress balance computation are minor when compared to those differences in the thermal computation. We agree that in complex regimes (high friction and large gradients in stress and strain), a higher vertical resolution should be better at capturing features associated with the stress balance. From our stress balance experiment, the larger differences between the 5 and 25-layer model tend to occur in these complex environments. Given the nature of our paper, and the benefit of using higher order vertical finite elements to improve the speed of model run targeted for paleoclimate experiments, we conclude that the stress balance is captured well when compared with the thermal computation, which relies on higher order vertical finite elements to achieve a similar accuracy as a model with a higher vertical resolution.

Following both Reviewer #1 and #2 suggestions we adjust (in revised version) Line 115 from:

"Although the stress balance computation does not require a high vertical resolution, the thermal model usually does in order to capture sharp thermal gradients near the base of the ice."

to (additional changes in the color red):

The majority of the computational demand for an ice sheet model resides within the stress balance computation. Although the thermal model requires many vertical layers in order to capture sharp thermal gradients near the base of the ice, stress balance tests performed with ISSM (not shown here) on models with 25 layers and 5 layers show the area averaged differences in the surface and basal velocities to be 0.22 % and 0.012% respectively. Therefore, for the purposes of the experiments outlined in this study, we consider that the stress balance computation does not require a high vertical resolution. As a consequence of the high number of vertical layers needed for the thermal computation, however, more runtime is needed during the stress balance computation than is necessary.

**Comment 3**: "Sec. 2.4. Figure 2 compares an exponential function captured by vertical elements with different polynomial interpolation. We understand that the figure is for 1 P3 element (i.e. 4 layers of nodes) or 3 P1 elements (i.e. 4 layers of nodes); but what is the corresponding number of P2 elements?"

**Response:** It seems the confusion surrounding Figure 2 is a mistake on our part regarding the wording of the caption. The figure on the left shows 3 prismatic elements, and on the right, we show the exponential profiles captured by the different finite elements for these 3 elements. We have changed the wording of the figure caption,

From:
"On the left is an example of P1xP3 prismatic elements. On the right is an example of exponential profile captured by P1, P2 and P3 finite elements. With higher order finite elements in the vertical, sharp gradients in temperature are captured more precisely than with a linear (P1) interpolation."

To:

"On the left is an example of 3 prismatic elements used to capture an exponential profile. On the right is an example of exponential profile captured by P1, P2 and P3 finite elements. With

higher order finite elements in the vertical, sharp gradients in temperature are captured more precisely than with a linear (P1) interpolation."

**Comment 4**: "Sec. 3.1: it is said that P1 elements are used for the stress balance. What is the default number of integration points. Does it allow to capture the temperature profile, affecting the viscosity, within the element?"

**Response:** We use Gauss-Legendre integration points in the vertical that capture polynomials of degree up to 9, which is adequate for the integrals that we have here.

**Comment 5**: "Sec. 3.1. All the introduction is about using higher order models for the stress balance, however most of the EISMINT comparison is done with the Shallow ice model, and we learn this very late in the results section. It should be said here that the 100 000 years experiment is done with the SIA, justifying the comparison with EISMINT2, and that the BP model is used only to do 100 years relaxations."

**Response:** Both reviewers expressed a need for more clarity in defining what type of model was used. We have therefore made adjustments in section 3 and 3.1. In section 3 (Model description and experimental setup) we make clear the SIA was used in the single dome experiment and BP used in the steady-state solution (Text added below is in the color red).

"We first test the precision of the higher-order vertical interpolation using a simplified single dome ice sheet experiment that uses the SIA, following experiment A of the European Ice Sheet Modeling INiTiative (EISMINT2) experiments (Payne et al., 2000). We then apply a similar setup to a GrIS wide model, where the steady-state thermal solution is computed using the BP model. Specifics regarding model setup and the relevant experiments are discussed below."

In section 3.1 (second sentence), we add:

"We perform all of our single ice dome experiments using the SIA on models with horizontal grid resolution of 20 km x 20 km, with a model domain of 1500 km x 1500 km"

**Comment 6**: "Page 5, line 227. It is said that the elements are finer in areas of steep topography and ice flow gradients. I think that the refinement is based on the second derivatives, not the gradients, so elements are finer where changes in slope and ice flow gradients are high?"

**Response:** We have fixed the wording of the sentence to reflect this (changes in red). In the revised version it is on line 286.

"The GrIS wide model relies on anisotropic mesh adaptation, whereby the element size is refined as a function of surface elevation (Howat et al., 2014) and InSAR surface velocities from Rignot and Mouginot (2012), becoming finer in areas in regions where the second derivative of these two quantities is higher."

**Figure 1**

A

[Figure]

[Figure]

Area avg. = 0.22%
Max = 6.2%

**B**

[Figure]

Basal velocity 5 layer

m/yr

Basal velocity 25 layer

m/yr

[Figure]

Basal velocity % differene 25 -5 layer

Area avg. = 0.012%
Max = 2.6%

C

[Figure]

Shear (sfc.-base) 5 layer    m/yr

Shear (sfc.-base) 25 layer    m/yr

[Figure]

Shear differene 25 - 5 layer    m/yr

Area avg. = 0.54 m/yr
Max = 87 m/yr

---

## Author Comment (AC2) · 5 Apr 2018

Reviewer #2 comments and response:

We thank the reviewer for his/her helpful comments and corrections. Below are responses to the particular comments and corrections.

**Comment 1:** "Introduction) It would be helpful to expand on the descriptions of the types of physics in order to more strongly make the case that Blatter-Pattyn is both desirable and expensive. I don't think you need any equations, but the hierarchy could be more explicit in terms of physical assumptions and their mathematical/numerical consequences. For a modeling journal like GMD, more detail than for a "regular" journal seems appropriate."

**Response:** We have adjusted the paragraph (in the revised version, lines 93-115) following the reviewer's recommendation (addition to prior text is in the color red) as follows:

With model-data comparisons of past ice sheet changes becoming more common, however, some applications may benefit from using an ice sheet model of increased complexity, particularly when comparisons of past margin behavior is of interest. Ideally, full stokes (FS) models provide a comprehensive 3D solution to the diagnostic equations. FS models, however, are prohibitively expensive computationally and are mainly relegated to modeling experiments no more than a few hundred years. As described above, SIA models represent the highest degree of simplification of the FS equations, in which the vertical shear stress is the only non-zero stress component in the force balance equations. Although advantageous due to its computational efficiency, SIA models cannot simulate ice streams, grounding line dynamics, and floating ice shelves. On the contrary, shelfy-stream or shallow-shelf approximation models (SSA; MacAyeal, 1989) were developed to be implemented in ice shelve regions where longitudinal stresses dominate, however, these models cannot represent slow flow in the interior of the ice sheet where vertical shear is non negligible. Higher-order models (Blatter, 1995; Pattyn, 2003; herein referred to as BP for Blatter-Pattyn) on the other hand, that include membrane stresses and elements of the vertical shear stress have been a hallmark in the ice sheet modeling community over the past decade, being favored for their ability to model both the fast and slow components of ice flow, while being computationally cheaper than FS models.

**Comment 2:** Line 96) "stress balance computation does not require a high vertical resolution" could use a reference."

**Response:** Because this is not a well-established metric, there is no known reference for the original statement made. However, from our experience with ISSM, users have found that the number of layers does not directly hinder an accurate stress balance computation.

In order to make this point without overstepping the boundaries of the existing literature, we refer the reviewer to the answer for **comment 8** and **comment 2 from reviewer #2**. Based on these tests we have changed our text in the introduction (revised version, lines 115 -122) to read (additional text in the color red):

"The majority of the computational demand for an ice sheet model resides within the stress balance computation. Although the thermal model requires many vertical layers in order to capture sharp thermal gradient near the base of the ice, stress balance tests performed with ISSM (not shown here) on models with 25 layers and 5 layers show the area averaged differences in the surface and basal velocities to be 0.22 % and 0.012% respectively. Therefore, for the purposes of the experiments outlined in this study, we consider that the stress balance computation does not require a high vertical resolution. As a consequence of the high number of vertical layers needed for the thermal computation, however, more runtime is needed during the stress balance computation than is necessary."

**Comment 3:** "Section 2) This could use a reference to your favorite finite element method book for the definitions of the elements."

**Response:** We have added a reference and sentence (revised version, line 224):

For more information about the finite element method, we direct the reader to Zienkiewicz and Taylor (1989).

Zienkiewicz, O.C. and Taylor R.L. The finite element method. Vol. I. Basic formulations and linear problems. London: McGraw-Hill, 1989. 648 p. Vol. 2. Solid and fluid mechanics: dynamics and non-linearity. London: McGraw-Hill, 1991. 807 p. [School of Engineering, University of Wales. Swansea, Wales]

**Comment 4:** "201) Totally OK to only run one of the EISMINT2 experiments, but explain very briefly why you picked A. Presumably because it's the initial spin-up?"

**Response:** We ended up choosing experiment A to help determine how our different set of experiments would perform under the initial relaxation, rather than for the remaining EISMINT 2 experiments which primarily tested the response of each model to various changes in forcings. Because we were concerned with the outcomes of the experiments using different vertical resolutions and difference vertical finite elements, our interest was in the spread of the various experiments in ISSM compared to the suite of models in the EISMINT2 experiment A relaxation.

To clarify in section 3.1, we added: Rather than performing all of the experiments associated with the EISMINT2 benchmarks, we choose to limit the analysis to only experiment A, where models begin from the same initial state.

**Comment 5:** "Section 3.1) Although you do say you're comparing to an experiment for SIA models, it should be more clear in the text that you're using SIA."

**Response:** Both reviewers expressed a need for more clarity in defining what type of model was used. We have therefore made adjustments in section 3 and 3.1. In section 3 (Model description

and experimental setup) we made clear the SIA was used in the single dome experiment and BP used in the steady-state solution (Text below in the color red is what was added).

"We first test the acuracy of the higher-order vertical interpolation using a simplified single dome ice sheet experiment that uses the SIA, following experiment A of the European Ice Sheet Modeling INiTiative (EISMINT2) experiments (Payne et al., 2000). We then apply a similar setup to a GrIS wide model, where the steady-state thermal solution is computed using the higher order BP model. Specifics regarding model setup and the relevant experiments are discussed below."

In section 3.1 (second sentence), we added:

"We perform all of our single ice dome experiments using the SIA on models with horizontal grid resolution of 20 km x 20 km, with a model domain of 1500 km x 1500 km"

**Comment 6:** "Section 3.2, first paragraph) It should be more clear what your thermal steady-state computation is, i.e. a simultaneous solution of heat transport and momentum equations coupled by viscosity. Also, the description of the effective viscosity can probably just say it's a function of strain rates/velocity gradients and a temperature dependent hardness. The variable name B isn't necessary when the equation isn't shown."

**Response:** The thermal steady state computation is done iteratively (e.g. thermal computation, stress balance, thermal computation, stress balance, etc.) until a user defined criterion is met (in this case, fixed at 10 iterations).

For the effective viscosity, we have decided to add the equation (line 275 in revised version):

$$\mu = \frac{B}{2\dot{\epsilon}_e^{\frac{n-1}{n}}}$$

"Where B is the ice hardness, n is Glen's flow law exponent and $\dot{\epsilon}_e$ is the effective strain rate. The ice hardness, B, is temperature dependent following the rate factors given in Cuffey and Paterson (2010, p. 75), while basal drag is empirically determined following a viscous flow law outlined in Cuffey and Paterson (2010)."

**Comment 7:** "Section 4.1) It's worth pointing out that Experiment A is a comparison of models without a known analytic solution, some of which produced more plausible (to me, anyway) solutions than others (e.g., W, X and Y). Comparison to the EISMINT2 results that don't show stability problems may be more reasonable. Also, it seems possible that your 25-layer results are better than the EISMINT2 models."

**Response:** We agree that no known solutions exist for the EISMINT2 experiments, and therefore have added the following sentence, line 335 in revised version: "It is important to note that no

known analytic solution was provided in the EISMINT 2 experiment A comparison." And we have adjusted the sentence following to read: "Similar to Rutt et al. (2009), however, we compare our simulated values to the mean and the standard deviation of the values for experiment A in the EISMINT2 experiment to assess the relative spread."

The reviewer brings up a good point that comparison to all models may not be necessary as many of the models experienced a thermal instability in the radial symmetry of the basal temperature. We have made a table (similar to Table 1 – attached below) to show how our simulations compare to the EISMINT2 experiment A models **W, Y, and Z**, which had a radially symmetric basal temperature during the spinup procedure. In general, our conclusions discussed in the paper remain the same, which leads us to favor including all models associated in the EISMINT2 experiment A rather than only using models **W, Y, and Z**. The conclusion is that when using fewer vertical layers, those models using the higher order vertical finite elements tend to match the model mean for the EISMINT2 experiment A models. This general conclusion is similar when using the mean of all models as well as the mean of only models **W, Y, and Z**.

**Comment 8**: "Section 4.2) Given that a major selling point of the new method is making BP affordable, I'd like to see some direct description of the dynamical results at some point. How do your calculated velocities compare for different numbers of layers and vertical elements? Is vertical shearing captured well? You only discuss experiments for which SIA is reasonable. I have some concern about whether the reduction in the number of levels will work for transient runs with more realistic geometry than the ice dome that include areas in which BP is necessary."

**Response:** This comment echoes concerns also shared from **Reviewer #1.** We are attaching the same analysis that was used to respond to **Reviewer #1 comment 2** (Figure 1. A, B, C)**.**

To test how sensitive the stress balance computation was in our GrIS model to changes in vertical resolution, we began by collapsing our 25 layer P1 model. By collapsing the 25 layer model, the ice viscosity parameter (B) is depth averaged, and therefore does not depend on depth. We next extruded our collapsed model to 25 layers and 5 layers, and ran a stress balance computation.

We have attached the results of that experiment (Figure 1. A, B, C). When comparing the stress balance surface velocity differences (A) between the 25 layer and 5 layer model, the area averaged difference is 0.22%, with the maximum difference being 6.2%. The area averaged difference in basal velocities (B) are 0.012%, with a maximum difference of 2.6%. Lastly, the area averaged difference in basal shear (C) is 0.54 m/yr with a maximum of 87 m/yr. From this experiment we conclude that the differences associated with the vertical resolution and the stress balance computation are minor when compared to those differences in the thermal computation. We agree that in complex regimes (high friction and large gradients in stress and strain), a higher vertical resolution should be better at capturing features associated with the stress balance computation. From our stress balance experiment the larger differences between the 5 and 25-

layer model tend to occur in these complex environments. Given the nature of our paper, and the benefit of using higher order vertical finite elements to improve the speed of model run targeted for paleoclimate experiments, we conclude that the stress balance is captured well when compared with the thermal computation, which relies on higher order vertical finite elements to achieve a similar accuracy as a model with a higher vertical resolution.

Although our tests show that the stress balance velocities are captured well using 5 layers, we understand that in transient experiments this may change. This is a larger question that we strongly regard to be acceptable for a different study separate than this one.

**Comment 9:** "347) "we began by using the relaxed model simulations that have thus far only used the shallow ice approximation. . ." Please be a little more clear that you mean the single dome calculations from 3.1/4.1."

**Response**: To clarify, we add: "we began by using the relaxed model simulations that have thus far only used the shallow ice approximation for the single dome experiments in section 4.1"

Technical corrections:

Line 76: Made the requested change and cut "a"
Line 77: Made the requested change and cut "for"
Line 80: Made the requested change from "towards" to "to"
Line 109: Made the requested change and added "more"
Line 129: Made the requested change and added a multiplication sign "×"
Line 143: Made the requested change and added spaces before and after the "=" sign.
Line 177: Made the requested change from "As" to "While"
Line 192: Made the requested change from "was" to "were", and changed "thermomechanically" to "thermomechanical"
Line 256: Made the requested change to "with respect to both"
Line 263: Made the requested change and cut "and above"
Line 357: Made the requested change from "criteria" to "criterion"

Table 2: We removed the units from the header.
Figure 3: We changed the last sentence of the figure caption from "Only those models that fall within 2% of the simulated ice volume for the 25 layer P1 model are labeled" to "Only those models that fall within 2% of the simulated ice volume for the 25 layer P1 model are labeled and colored as shown in their respective legends"

**Table 1 using EISMINT2 models W,Y, and Z**

| | Volume ($10^6 \text{km}^3$) | Ice divide basal temp (K) | Ice divide thickness (m) |
|---|---|---|---|
| **Eismint 2 exp. A (mean value)** *Payne et al., 2000* | **2.134 ± 0.03** | **256.28 ± 0.80** | **3676.5 ± 55.65** |
| 25 layer P1 | 2.144 | 254.723 | 3767.0 |
| 3 layer P1 | 2.344 | 247.229 | 4093.2 |
| 4 layer P1 | 2.265 | 250.240 | 3960.4 |
| 5 layer P1 | 2.231 | 252.351 | 3876.5 |
| 6 layer P1 | 2.209 | 253.285 | 3844.4 |
| 7 layer P1 | 2.192 | 253.793 | 3823.0 |
| 8 layer P1 | 2.179 | 254.115 | 3806.7 |
| 9layer P1 | 2.171 | 254.337 | 3794.5 |
| 10 layer P1 | 2.165 | 254.480 | 3785.4 |
| 3 layer P2 | 2.264 | 249.873 | 4023.2 |
| 4 layer P2 | 2.169 | 252.598 | 3838.1 |
| 5 layer P2 | 2.146 | 253.717 | 3785.8 |
| 6 layer P2 | 2.138 | 254.225 | 3764.8 |
| 7 layer P2 | 2.131 | 254.488 | 3753.9 |
| 8 layer P2 | 2.124 | 254.532 | 3747.1 |
| 9 layer P2 | 2.123 | 254.634 | 3743.6 |
| 10 layer P2 | 2.122 | 254.656 | 3741.3 |
| 3 layer P3 | 2.245 | 250.019 | 4002.0 |
| 4 layer P3 | 2.160 | 252.689 | 3826.4 |
| 5 layer P3 | 2.145 | 253.581 | 3779.3 |
| 6 layer P3 | 2.143 | 253.895 | 3765.0 |
| 7 layer P3 | 2.138 | 254.213 | 3756.5 |
| 8 layer P3 | 2.131 | 254.334 | 3750.3 |
| 9 layer P3 | 2.129 | 254.436 | 3748.5 |
| 10 layer P3 | 2.127 | 254.600 | 3746.2 |

Table 1 - **Reviewer 2 comments**.  Ice volume, ice divide basal temperature, and ice divide thickness for each indivicual simulation after 100 kyr.  Also shown is the corresponding values for the EISMINT2 (Payne et al., 2000) experiment A simulations **W, Y, Z**, which were noted in Payne et al. (2000) to produce a more radially symmetric basal temperature.  The shading indicates those simulations whose values fall within 1 standard deviation (green), 2 standard deviations (blue,) and 3 standard deviations (red) from the EISMINT2 mean values for experiments W, Y, Z.

A

[Figure]

Surface velocity 5 layer    m/yr

Surface velocity 25 layer    m/yr

[Figure]

Surface velocity % differene 25 -5 layer

Area avg. = 0.22%
Max = 6.2%

**B**

[Figure]

Basal velocity 5 layer — m/yr

Basal velocity 25 layer — m/yr

[Figure]

Basal velocity % differene 25 -5 layer

Area avg. = 0.012%
Max = 2.6%

C

[Figure]

Shear (sfc.-base) 5 layer — m/yr

Shear (sfc.-base) 25 layer — m/yr

[Figure]

Shear differene 25 -5 layer — m/yr

Area avg. = 0.54 m/yr
Max = 87 m/yr